# Tumor-Promoting ATAD2 and Its Preclinical Challenges

**DOI:** 10.3390/biom12081040

**Published:** 2022-07-28

**Authors:** Haicheng Liu, Qianghai Wen, Sheng Yan, Weikun Zeng, Yuhua Zou, Quanliang Liu, Guoxi Zhang, Junrong Zou, Xiaofeng Zou

**Affiliations:** 1Department of Urology, First Affiliated Hospital of Gannan Medical University, Ganzhou 341000, China; liuhaicheng0762@163.com (H.L.); gnyxywqh@163.com (Q.W.); yansheng20202194@163.com (S.Y.); z1990819817@163.com (W.Z.); gnugz808@163.com (Y.Z.); liuquanliang2008@163.com (Q.L.); zgx8778@163.com (G.Z.); 2The First Clinical College of Gannan Medical University, Ganzhou 341000, China; 3Institute of Urology, Gannan Medical University, Ganzhou 341000, China; 4Jiangxi Engineering Technology Research Center of Calculi Prevention, Gannan Medical University, Ganzhou 341000, China

**Keywords:** ATAD2, malignant tumor, bromodomain, pharmacology

## Abstract

*ATAD2* has received extensive attention in recent years as one prospective oncogene with tumor-promoting features in many malignancies. ATAD2 is a highly conserved bromodomain family protein that exerts its biological functions by mainly AAA ATPase and bromodomain. ATAD2 acts as an epigenetic decoder and transcription factor or co-activator, which is engaged in cellular activities, such as transcriptional regulation, DNA replication, and protein modification. ATAD2 has been reported to be highly expressed in a variety of human malignancies, including gastrointestinal malignancies, reproductive malignancies, urological malignancies, lung cancer, and other types of malignancies. ATAD2 is involved in the activation of multiple oncogenic signaling pathways and is closely associated with tumorigenesis, progression, chemoresistance, and poor prognosis, but the oncogenic mechanisms vary in different cancer types. Moreover, the direct targeting of ATAD2’s bromodomain may be a very challenging task. In this review, we summarized the role of ATAD2 in various types of malignancies and pointed out the pharmacological direction.

## 1. Introduction

Malignancy is one of the most common diseases, a process consisting of complex and continuous changes, the incidence of which is increasing yearly and is the second leading cause of death worldwide [1,2]. Recent studies have revealed that ATPase family AAA domain-containing 2 (ATAD2) is overexpressed in various human malignancies [3]. There are two isoforms of human *ATAD2*, known as *ATAD2A* and *ATAD2B* [4], and most published functional studies have been done on ATAD2/ATAD2A; hence, ATAD2 is also the focal point of our review. ATAD2 is mainly expressed in male genital cells [5] and has two conserved protein domains, AAA ATPase and bromodomain (BRD), which are the main structures for its biological functions [6,7]. Studies have shown that ATAD2 acts as an epigenetic decoder and transcription factor or coactivator involved in many cellular processes, such as DNA replication [8], transcriptional regulation [9], protein modification [10], and cell proliferation [11]. ATAD2 overexpression and downstream signaling pathways affect the biological function of cells in a variety of different ways, contributing to the carcinogenic process. ATAD2 first attracted attention as a co-activator of estrogen and androgen receptors in breast cancer [12,13], and subsequent research also found a high expression of ATAD2 in other human cancers [3]. In a novel human pluripotent stem cell-based cancer model, ATAD2 is required for the response to the oncogene BRAF and tumor initiation in melanoma. ATAD2 is a key chromatin modifier that forms a complex with SOX10, enabling the expression of downstream oncogenic programs and promoting the melanoma phenotype [14]. It is currently believed that high ATAD2 expression is associated with high histological grade, low overall survival, and tumor metastasis and recurrence in a variety of malignancies [15], but the ATAD2 oncogenic signaling pathway has not been fully clarified.

The overexpression of ATAD2 in various human malignancies causes the imbalance in proliferation and apoptosis of tumor cells leading to tumor development (Table 1). Although the current research has made some progress, the main mechanism of ATAD2 is not the same for different types of malignant tumors. We provide a comprehensive review of ATAD2, focusing on its structure, biological functions, and roles in various malignancies.

## 2. ATAD2 Is a Member of Bromodomain-Containing Proteins

The bromodomain is an evolutionarily highly conserved protein interaction module that recognizes lysine acetylation motifs [62], which are key events in the reading of epigenetic marks, and it was also later found that BRD can bind propionylated and butyrylated lysine residues [63]. There are about 42 identified human BRD-containing proteins, each containing between one and six BRDs [64]. Bromodomain-containing proteins have multiple functions (Figure 1), including transcriptional regulation, chromatin remodeling, and histone modifications. A variety of bromodomain proteins have been observed to be overexpressed and play a role in malignant tumors. Bromodomain proteins have been classified into eight subfamilies based on their sequence and structural similarities, and ATAD2 belongs to family IV bromodomain proteins [65,66]. *ATAD2*, also known as *ACNNA*, has a chromosomal location of 8q24.13, which is a commonly amplified region in cancer [67,68,69]. *ATAD2* is composed of 28 exons and its protein contains 1390 amino acids at 158.5 kDa molecular mass [70]. ATAD2 can be divided into four regions, N-terminal acidic domain, AAA ATPase domain (AAA-ATAD2), bromodomain, and C-terminal domain (Figure 1). AAA-ATAD2 contains Walker A, Walker B, sensor 1, sensor 2, and an arginine finger and functions as a hexameric compound [7]. The bromodomain has four α-helix bundles and ZA and BC loops between the two helices, which contain several amino acids necessary to interacting with acetylated lysines to form a hydrophobic pocket [71]. Both the AAA ATPase and bromodomain are the most protected domains of ATAD2 [9] and are also the main targets of cancer drug therapy research [3,67]. *ATAD2B* is the human homolog gene of *ATAD2*, but little research has been done on ATAD2B. ATAD2 and ATAD2B are highly conserved and similar, they both contain an AAA domain and a bromodomain with amino acid sequence homology of 97% and 74%, respectively. ATAD2B expression in the vertebrate nervous system is transiently expressed in the nucleus of developing neuronal cells, suggesting that it has acquired more cell-specific roles. The expression of ATAD2B in human tumors was also investigated. Oncomine and immunohistochemistry showed a high expression of ATAD2B in glioblastoma and oligodendroglioma; ATAD2B immunostaining was also increased in human breast cancer. In tumors, ATAD2B appears to be cytoplasmic or membrane bound, rather than nuclear. ATAD2B may play a role in neuronal differentiation and tumor progression, but further studies are needed [72].

## 3. The Biological Function of ATAD2

### 3.1. Transcriptional Regulation

Transcription is a highly regulated innate random biochemical process [73] that is completed by binding to specific regions of DNA and RNA [74]. ATAD2 utilizes the energy released from the hydrolysis of ATP by the ATPase domain to reconfigure the partial chromatin architecture, which is a critical stage of transcription [75]. Studies have indicated that Yta7, an ATAD2 homolog in *Saccharomyces cerevisiae*, is a possible activator of histone gene transcription [76,77]. The binding histone gene HTA1 by Yta7 is accurately regulated in the cell cycle. After Yta7 recruits RNA polymerase II to histone genes, multiple sites at the N-terminal of Yta7 are phosphorylated with at least two diverse kinases, CDK1 (cyclin-dependent kinase 1) and CK2 (casein kinase 2), and phosphorylated Yta7 is freed by HTA1, a process that is closely related to the optimal transcription of HTA1, as well as additional histone genes [78]. Subsequent studies further demonstrated that ATAD2 acts as an epigenetic reader during DNA transcription and its bromodomain recognizes and binds histone acetylated lysines. It should be noted that this process proceeds in an acetylation-independent manner [79,80]. ATAD2 is then recruited to specific promoters of target genes and is involved in the recruiting or assembling of transcriptionally active complexes of proteins (including CBP) on target genes and the histone modifications mediated by them to promote target gene transcription [12,13,81]. It has also been recently reported that ATAD2 can promote cell differentiation and proliferation by serving as a pluripotent enhancer of chromatin dynamics [6]. ATAD2 is often considered a critical transcription factor or coactivator among malignant tumor cells [13,82], regulating the transcriptional levels of related genes, which in turn promotes the proliferative and apoptotic activities of tumor cells through multiple signaling pathways, which require the combined action of the ATPase domain and the bromodomain [3,67,83].

### 3.2. DNA Replication

DNA is a double-helix structure formed by complementary base pairing of deoxynucleotides and serves as the storage and transmission material of the main genetic information of organisms [84]. DNA replication is the central link of cell proliferation, and genomic DNA replication can generally be divided into three stages: (1) Initiation, in which the origin of DNA replication is unwound by the replicative DNA helicase. (2) Elongation, in which forks copy the chromosome using semi-conservative DNA synthesis. (3) Termination, when converging replication forks meet [85,86]. Studies have revealed that ATAD2 is recruited to the replication site during the replication initiation phase by interacting with newly synthesized histones, a mechanism involving the formation of diacetylation marks at K5 and K12 by newly synthesized histone H4 at the replication site. A conserved asparagine in the hydrophobic pocket of the ATAD2 bromodomain interacts directly with diacetylated lysines, a phenomenon that occurs transiently during the recombination of replication-coupled nucleosomes [71,87,88,89]. Morozumi et al. revealed the presence of a large number of key DNA replication factors associated with nucleosome-bound ATAD2 [6]. Furthermore, ATAD2 appears to be more associated with heterochromatin replication, during which ATAD2 expression levels are increased and localized to the sites of heterochromatin replication through physical interactions with heterochromatin components [90].

### 3.3. Other Functions

The biological functions of ATAD2 extend far further than this. Similar to most AAA+ superfamily members, the ATPase domain of ATAD2 has an important role in the oligomerization of proteins and is in charge of the binding and hydrolysis of ATP. ATAD2 hydrolyzes ATP by the nucleophilic attack on phosphorylated adenosine, breaking high-energy phosphate bonds, converting ATP into ADP and free inorganic phosphate, and releasing a large amount of energy [91,92]. The energy provided promotes various cellular processes necessary for life (Figure 1), including protein folding [93], intracellular transport [94], protein degradation [95], DNA repair [96], DNA remodeling [97], and ion transport [98], which has broad biological implications. Notably, ATAD2 is usually required to assemble into hexameric ring complexes during the energy-dependent remodeling of biomacromolecules in cells [99]. In addition, ATAD2 plays an essential role in the formation of higher-order chromatin structures. ATAD2 hexamers can act as scaffolding proteins that guide the construction of higher-order chromatin structures by bringing nucleosomes into close proximity, and the formation of higher-order chromatin structures requires tight regulation of acetylation and deacetylation, a process that ATAD2 may regulate by competing with histone deacetylase 1 (HDAC1) [90,100]. ATAD2 also controls the dynamics of histone chaperone and chromatin interactions, particularly the histone chaperone HIRA. ATAD2 deletion would significantly reduce histone chaperone and chromatin interactions, causing an accumulation of histone chaperones at the active gene transcription start site, which leads to a disruption of the nucleosome assembly–disassembly balance, resulting in a net increase in nucleosome assembly [101].

## 4. The Role of ATAD2 in Human Malignant Tumors

### 4.1. ATAD2 in Digestive System Malignant Tumors

#### 4.1.1. ATAD2 in Esophageal Cancer (EC)

With the rapidly increasing morbidity of esophageal cancer, disease-related molecular events have also attracted the attention of researchers [102]. Liu et al., found that the ATAD2 gene copy number is increased in primary small cell esophageal carcinoma (SCEC) and may play an important role in SCEC through WNT and NOTCH signaling pathways [16]. In esophageal squamous cell carcinoma (ESCC) tissues and cell lines, ATAD2 is highly expressed, which relates to tumor TNM stage and clinicopathological progression and accelerates ESCC development through the Hedgehog (HH) signaling pathway [17,18]. Cao et al., found that ESCC exerts pro-metastatic effects through the C/EBPβ/TGF-β1/Smad3/Snail signaling pathway (Figure 2). ATAD2 interacts directly with C/EBPβ to promote their nuclear translocation, and C/EBPβ combines directly with the TGF-β1 promoter region to activate its expression. TGF-β1 activates its downstream effector molecules in a Smad3-dependent fashion. Moreover, ATAD2 also accelerates ESCC metastasis through TGF-β1 signaling that induces Snail expression and subsequent epithelial–mesenchymal transition (EMT). These discoveries revealed a fresh molecular strategy for the function of ATAD2 in ESCC and identify a hopeful target for the treatment of ESCC patients [19].

#### 4.1.2. ATAD2 in Gastric Cancer (GC)

HIF1α binding sites (HBS) and HIF1α auxiliary sites (HAS) were discovered in the *ATAD2* promoter and that HIF1α combines with the *ATAD2* promoter under hypoxic conditions to enhance *ATAD2* expression [20]. ATAD2 was involved in the pRb-E2F1 signaling pathway in GC (Figure 2). ATAD2 positively regulates the expression of key cell cycle regulatory proteins, including cyclinD1, ppRb, E2F1, and cyclinE. The pRb is a CDK4/CDK6 target, and as soon as cyclin D1 combines with CDK4/CDK6, pRb is phosphorylated and ppRb is liberated from E2F. ATAD2 binds to the transcription factors E2F and c-MYC to promote proliferation-related and anti-apoptotic gene expression, causing the occurrence and development of GC [21,22]. Furthermore, ATAD2 may interact with estrogen receptor 1 (ESR1) to regulate nuclear receptor coactivator 1 (NCOA1) and protein arginine methyltransferase 1 (PRMT1) to cause epigenetic changes in GC (Figure 2) [23]. The top-ranked partners binding to ATAD2 were identified by protein–protein interactions (PPI) in GC; ESR1, SUMO2, SPTN2, and MYC preferred bromodomain, while NCOA3 and HDA11 favored the ATPase domain of ATAD2. Insight into the ATAD2–PPI interface provides a new target for gastric cancer therapy [103]. Peritoneal metastasis occurs in 40–60% of patients with gastric cancer recurrence after surgical treatment, which greatly limits the patient’s quality of life. Intraperitoneal chemotherapy with paclitaxel is an effective method for the treatment of gastric cancer with peritoneal metastasis [104,105]. It has been found that *ATAD2* can be used as one of the candidate genes for paclitaxel resistance and become a potential new marker for predicting paclitaxel resistance in patients with peritoneal metastasis of GC [106]. ATAD2 overexpression is correlated with the clinical stage, depth of tumor invasion, lymph node, and distant metastasis of GC and is an independent factor in the prognosis of GC patients [21,107].

#### 4.1.3. ATAD2 in Hepatocellular Cancer (HCC)

ATAD2 is overexpressed in hepatocellular carcinoma, and its overexpression level positively correlates with aggressive phenotype and disease progression [108]. ATAD2 promotes the proliferation of liver cancer through the miR-520a/E2F2 pathway. ATAD2 acts directly on E2F2 and also negatively regulates miR-520a, which in turn increases E2F2 expression to promote hepatocellular carcinoma cell growth, and ATAD2 inhibits the expression of endoplasmic reticulum oxidoreductin 1 (ERO1L) and Ras-GTPase-activating protein-SH3-domain-binding protein 2 (G3BP2), which enhance the migration of hepatocellular carcinoma cells (Figure 2) [24]. In addition, ATAD2 can also cooperate with c-MYC to control the expression of smoothened (SMO) and glioma-associated oncogenes (GLI), and activate the Hedgehog signaling pathway. Long non-coding RNA (lncRNA) PCAT-14 inhibits miR-372 to promote ATAD2 expression in the activated Hedgehog pathway (Figure 2). ATAD2 also negatively regulates the expression of APC that inhibits β-catenin and accelerates the occurrence of HCC, and ATAD2 promotes HCC progression by inhibiting the p53/p38-mediated apoptosis signaling pathway (Figure 2) [25,26,27,28,29]. ATAD2 can positively regulate kinesin family member 15 (KIF15), which further promotes HCC stem cell phenotype and malignancy through reactive oxygen species imbalance (ROS) [30]. ATAD2 is a potential proliferation marker for liver regeneration and HCC, but there is a high degree of heterogeneity in the effect of ATAD2 inactivation on gene expression in different HCC cell lines [109]. Studies have shown that ATAD2 has important diagnostic and prognostic value in HCC patients [110,111,112].

#### 4.1.4. ATAD2 in Pancreatic Cancer (PC)

Pancreatic cancer is one of the most invasive human tumors with a poor prognosis. ATAD2 deletion inhibited the invasive and migratory functions of pancreatic cancer cells (PCCs) and made them susceptible to gemcitabine, and ATAD2 gene knockout inhibited the non-anchored growth of PCCs in vitro [31,113]. The results indicate that ATAD2 is responsible for the malignant features of pancreatic cancer. Dutta et al., discovered a potential binding site for miR-217 in the 3′UTR of *ATAD2*, and then confirmed by luciferase analysis that *ATAD2* is a direct target of miR-217. The miR-217 overexpression significantly downregulated ATAD2 expression in PCCs and inhibited the proliferation and migration and induced apoptosis and cell cycle arrest in PCCs. During, miR-217 blocked pancreatic cancer progression by inactivating the AKT pathway, which may be partly due to miR-217 mediated inhibition of ATAD2 expression [32], but further studies are needed.

#### 4.1.5. ATAD2 in Colorectal Cancer (CRC)

Research has found that ATAD2 is highly expressed in CRC [114]. Like gastric cancer, ATAD2 also promotes the proliferation of CRC cells through the pRb-E2F1 pathway [33]. As a coactivator of E2Fs, ATAD2 also promotes the expression of oncogenic ubiquitin E3 ligase (TRIM25). TRIM25 can interact with ATAD2 and stabilize its resistance to genotoxic injury, thus forming an ATAD2-E2Fs-TRIM25 positive feedback loop that drives CRC progression [34]. In addition, ATAD2 is involved in CRND-mediated miR-126-5p/ATAD2 axis. CRNDE directly binds and inhibits the target gene miR-126-5p that negatively regulates the expression of ATAD2, thereby promoting the development of colorectal cancer and paclitaxel resistance. In addition, ATAD2 enhanced VEGFA secretion by inhibiting miR-520a to promote CRC angiogenesis (Figure 2) [35,36]. ATAD2 overexpression is linked to colorectal progression and prognosis [115].

### 4.2. ATAD2 in Reproductive System Malignant Tumors

#### 4.2.1. ATAD2 in Ovarian Cancer (OC)

Wrzeszczynski et al., identified *ATAD2* as an oncogene with overexpressed and hypomethylated properties in ovarian cancer [116]. ATAD2 is both a marker and driver of cell proliferation in OC. ATAD2 may drive OC cell proliferation through the MAPK pathway and is also involved in the process of miR-372 inhibiting OC cell proliferation [37,38]. ATAD2 participates in the highly activated PI3K/AKT oncogenic channel, while miR-200b-5p targets the inhibition of ATAD2 expression and regulates the PI3K/AKT pathway to suppress proliferation and accelerate apoptosis in OC cells [39]. The resistance of ovarian cancer chemotherapy patients to platinum drugs is a problem that cannot be ignored, which is related to the miR-302/ATAD2 axis. ATAD2, a directed target of miR-302, inhibits APC and promotes nuclear β-catenin expression to reduce the sensitivity of ovarian cancer cells to cisplatin and enhance cisplatin-resistant cell migration, invasion, and EMT abilities [40]. Therefore, ATAD2 has important potential value for the treatment and prognosis of OC.

#### 4.2.2. ATAD2 in Uterine Corpus Endometrial Carcinoma (UCEC)

ATAD2 overexpression is associated with MYC expression and the amplification of the 8q24 region in UCEC [41]. High ATAD2 expression caused changes in cell cycle regulation and B-MYB-related genes, such as increased expression of B-MYB, E2Fs, and KIFs. These genes cooperate to drive cancer cell proliferation and enhance tumor aggressiveness [42]. ATAD2 is also positively correlated with UCEC stage, histological grade, depth of myometrial invasion, lymph node metastasis, lymphatic space involvement, and recurrence, and can be used as an independent poor prognostic indicator [117,118].

#### 4.2.3. ATAD2 in Cervical Cancer (CC)

Through bioinformatics analysis, ATAD2 takes an essential role in the pathogenesis of cervical cancer, and ATAD2 overexpression in cervical cancer promotes tumor cell proliferation, invasion, and metastasis. It can serve as a potential prognostic marker and therapeutic target for cervical cancer [43,119].

### 4.3. ATAD2 in Urinary System Malignant Tumors

#### 4.3.1. ATAD2 in Prostate Cancer (PCa)

The androgen receptor (AR) plays a key role in the mechanism of prostate carcinogenesis, and it exerts an important influence on prostate cancer cell proliferation, survival, and differentiation, mainly by regulating the androgen-induced different gene expression program [120,121]. ATAD2 also shows a high expression in numerous prostate cancer subtypes [13]. Guo et al., demonstrated the joint mediation of ATAD2 expression in prostate cancer by AR and E2F1 in the presence of androgens, and ATAD2 directly promotes the expression of the target gene NSD2. Histone methyltransferase NSD2 is a critical chromatin modulator of the NF-κB pathway and a regulator of the cytokine autocrine cycle, involved in the mobilization of NF-κB. The upregulation of NSD2 expression acts as a key function for prostate cancer cell proliferation, survival, and tumor angiogenesis, and ATAD2 acts as an AR co-activator that enhances its transcriptional activity [44]. ATAD2 with AR is convened to specific AR target genes in response to androgen stimulation, promoting their expression. The expression inhibition of ATAD2 intensely suppressed androgen-responsive or non-androgen-dependent AR-positive prostate cancer cell proliferation and led to a significant rise in tumor cell apoptosis. The expression of ATAD2 in human prostate surgical specimens was detected by immunohistochemistry and its outcome revealed that ATAD2 overexpression in prostate cancer was associated with disease progression [13].

#### 4.3.2. ATAD2 in Renal Cancer (RC)

ATAD2 was investigated to be highly expressed in renal cancer, and ATAD2 serves as a target of miR-372 directly in renal cancer cell lines and is regulated negatively by miR-372, which in turn affects cancer cell invasion, metastasis, and EMT function. ATAD2 is an attractive biomarker and therapeutic target for renal cancer [45,122].

### 4.4. ATAD2 in Respiratory System Malignant Tumors

#### ATAD2 in Lung Cancer (LC)

Lung cancer, as one of the most widespread malignancies, has a low survival rate, which makes it the primary cause of cancer-related deaths globally [1,123]. In recent years, the molecular biological characteristics of lung cancer have been extensively studied, and the identification of potential molecular targets has led to more and more lung cancer patients using molecular targeted therapy, which has effectively improved the survival rate of patients [124,125]. Variations in *ATAD2* in smokers were found in non-small cell lung cancer (NSCLC) patients, and *ATAD2* amplification not only regulates MYC-dependent transcription but is also a major driver of MYC-promoting lung adenocarcinoma cell proliferation [46]. Recently, the overexpression of ATAD2 was found to be positively correlated with the expression of maximum standardized uptake value (SUVmax), total lesion glycolysis (TLG), glucose transporter protein 1 (GLUT1), and hexokinase 2 (HK2) in lung adenocarcinoma (LUAD) tissues. The study further pointed out that this was caused by ATAD2 promoting LUAD glucose metabolism via the AKT-GLUT1/HK2 pathway. In addition, ATAD2 also promotes the proliferation, tumorigenicity, and migration of lung cancer cells utilizing the PI3K/AKT pathway [47]. *ATAD2* knockdown inhibited the migration, invasion, stem cell-like properties, and mitochondrial reactive oxygen species (mtROS) production of lung cancer cells. Whereas chronic intermittent hypoxia (CIH)-induced HIF-1α significantly activated the expression of *ATAD2*. The integrity of HIF-1α/ATAD2 triggered by CIH may determine the aggressiveness of lung cancer through the interaction of mtROS and stemness in lung cancer cells [48]. ATAD2 overexpression is closely associated with positive LUAD lymph node metastasis, poor tumor differentiation, advanced disease stage, and prognosis, and it is an individual marker of adverse prognosis after surgical resection of lung adenocarcinoma [5,47,126,127].

### 4.5. ATAD2 in Other Types of Malignant Tumors

It was also found that ATAD2 overexpression in other types of malignancies has multiple pro-tumor proliferation and survival roles (Figure 3). The high ATAD2 expression in breast cancer (BC) is caused by 8q24 amplification, which is closely related to the activation of multiple MYC pathways [41]. ATAD2 overexpression is associated with histological grade, tumor metastasis, and poor survival in BC, and it is a potential drug therapy target [128,129]. Its abnormal expression regulates multiple pathways of BC cell proliferation and survival. ATAD2 is involved in the pRB/E2F/c-MYC oncogenic signaling pathway in BC. ATAD2, a straight target of the pRB/E2F pathway, promotes tumorigenesis by binding to MYC/MYC target genes and cooperating with MYC for transcriptional activation [9]. ATAD2 also promotes breast cancer proliferation and survival through steroid hormone signaling. ATAD2 is strongly induced by E2, cooperates with E2F to positively regulate the proto-oncogene ACTR, and directly interacts with ERα and ACTR to stimulate the ERα target gene presentation, causing cell cycle progression (G1/S transition) and proliferation of breast cancer cells induced by estrogen [12,49,50]. ATAD2 also takes part in the PI3K-AKT-mTOR oncogenic signaling pathway in breast cancer [51,52]. ATAD2 was discovered to be a key factor in the deregulation of the kinesin family (KIFs), and the dysregulation of kinesin promotes cancer growth [53]. Moreover, ATAD2 is a critical intermediary of DNA damage response and repair in breast cancer cells, through mediating Chk1, Chk2, and BRCA1, which have an essential role in the dissolution of DNA damage foci and homologous recombination in BC [54]. ATAD2 is not only tumorigenic but also associated with immune cell infiltration in the tumor microenvironment in osteosarcoma (OS). ATAD2, a downstream target of methyltransferase-like 3 (METTL3), is positively regulated by METTL3 and is a biomarker for predicting prognosis and a candidate target for diagnosis and treatment of OS [55,130,131,132]. ATAD2 is highly expressed and directly negatively regulated by miR-186 in retinoblastoma (RB). ATAD2 activates the Hedgehog signaling pathway to promote RB cell viability, invasion, migration, and angiogenesis and is also involved in the lncRNA MALAT1/miR-655-3p/ATAD2 axis to accelerate RB progression [56,57]. ATAD2 overexpression was associated significantly with the expression of PD-L1, B7-H4, ALDH1, Slug, and CMTM6 proteins in oral squamous cell carcinoma (OSCC), suggesting that ATAD2 played an important function in the regulation of EMT, immunosuppression, and CSCs in OSCC [58]. ATAD2 promotes papillary thyroid cancer (PTC) progression through the lncRNA NEAT1_2/miR-106b-5p pathway [59]. Exogenous ATAD2 overexpression can significantly increase the expression of Polo-like kinase 4 (PLK4) to promote the occurrence and radiation resistance of glioblastoma (GBM), and so ATAD2 may be a key regulator of PLK4 transcription in GBM [60]. ATAD2 may be involved in the occurrence of nasopharyngeal carcinoma (NPC) by regulating the cell cycle and nucleic acid metabolism process, and it is also a molecular biomarker for the early diagnosis of NPC [61]. However, there are fewer studies of ATAD2 in these tumors and more clinical trials are still needed.

## 5. Preclinical Challenges for ATAD2

A protein is considered a “druggable” target when its activity can be regulated by a “drug” [133]. Given the important role of ATAD2 in malignancies, the exploration of small molecule inhibitors targeting ATAD2 has become one of the hot topics in oncology research. In the beginning, the predicted druggability of ATAD2 was low according to the data from computational analyses [134,135]. However, subsequent drug discovery works have demonstrated it to be a pharmacologically manageable target [136,137,138]. Currently, studies have identified several 3D structures characterizing the BRD of ATAD2, which have been successfully used for computational druggability assessment and drug discovery research [134,135]. A series of small molecule inhibitors against ATAD2’s BRD, such as thymidine nucleoside analogs [136] and quinolones [139], have been discovered by fragment-based approaches. However, these substances still need further structural optimization to reduce the equilibrium dissociation constants and hydrophilicity and to improve selectivity and activity. Later, GlaxoSmithKline ultimately identified compound 23 (GSK8814) by a structure-based approach, which was the first low-nanomolar, selective and cell-permeable chemical probe against the BRD of ATAD2, though the compound still suffered from a lack of activity [137]. A combination of structure-based virtual screening and biochemical analysis led to the finding of a new inhibitor (AM879) for ATAD2 bromodomain, which is able to trigger apoptosis and autophagy in breast cancer cells via PI3K-AKT-mTOR signaling [51]. Bamborough et al., first used high-throughput screening techniques to identify phenylsulfonamides as ATAD2 inhibitors. These compounds have a novel ATAD2 binding mode with atypical features, including the replacement of all conserved water molecules and halogen-bonding interactions within the active site. In addition, an orthogonal biophysical approach was used to optimize the hit identification strategy to identify individual active series [140]. On this basis, Lucas et al., performed the optimization of a series of phenylsulfonamides by displacing a conserved network of four water molecules, which exhibited a novel binding mode to non-bromodomains and extra terminal domain (non-BET) bromodomains. GSK388 was identified after continuous structural optimization with an ATAD2 inhibitor with excellent potency and a novel binding mode with a selectivity of more than 200-fold over the BET family. Although GSK388 still has residual activity against other bromodomains, it is a valuable complement to ATAD2 inhibitors [141]. Recently, Winter-Holt et al. also applied high-throughput screening techniques to identify a range of new ATAD2 inhibitors, confirming bromodomain as the site of action, and they performed optimization strategies to enhance the potency, selectivity, and permeability of the initial hit. The final screening resulted in the identification of compound 5 (AZ13824374), a potent and selective ATAD2 inhibitor. It showed cellular target engagement and antiproliferative activity in a series of breast cancer models [142]. Unfortunately, the available 3D structure for the AAA ATPase domain of ATAD2 was not discovered, which may be the main reason for the hindrance in the research of its small molecule inhibitors. Although some progress has been made in the research of drugs targeting ATAD2, numerous difficulties and challenges are still faced, and there is still a long distance from the clinical trials of the drugs.

## 6. Conclusions and Prospects

Malignant tumors are a major reason for human death, and their diagnosis and treatment have always been a major challenge to human medicine and an urgent problem to be solved. ATAD2 is a very promising tumor-promoting factor, and it is necessary to study its function in tumors.

ATAD2 is a member of the family IV bromodomain-containing proteins. Under the joint action of the conserved AAA ATPase domain and bromodomain, ATAD2 takes part in a range of cellular activities, such as transcriptional regulation, chromatin remodeling, histone modification, and the formation of higher-order chromatin structures. As an epigenetic reader, ATAD2 recognizes and binds mainly acetylated histone lysines through the bromodomain, which modulates the transcriptional activity of target genes. These genes promote the occurrence and development of malignant tumors by activating and participating in multiple oncogenic signaling pathways that enhance tumor cell proliferation, migration, invasion, and EMT, and they are associated with distant metastasis and chemotherapy resistance.

The close association between ATAD2 and malignancies suggests that ATAD2 has multifaceted oncogenic effects and is not only an emerging biomarker for the diagnosis and prognosis of many malignant tumors but also a potential drug target for the therapy of malignancies. However, the recognition mode of ATAD2 and histone targets and the oncogenic mechanism of it is undetermined. Although several small molecule inhibitors of ATAD2 have obtained some effects in cellular assays, more efforts are needed to develop specific inhibitors of ATAD2. In addition, several intracellular mediators that regulate ATAD2, such as long non-coding RNAs and microRNAs, and drugs targeting these mediators may also be a new option for tumor therapy. Therefore, ATAD2 is a very promising novel tumor-promoting factor with broad research prospects for the diagnosis and treatment of malignant tumors.

## Figures and Tables

**Figure 1 biomolecules-12-01040-f001:**
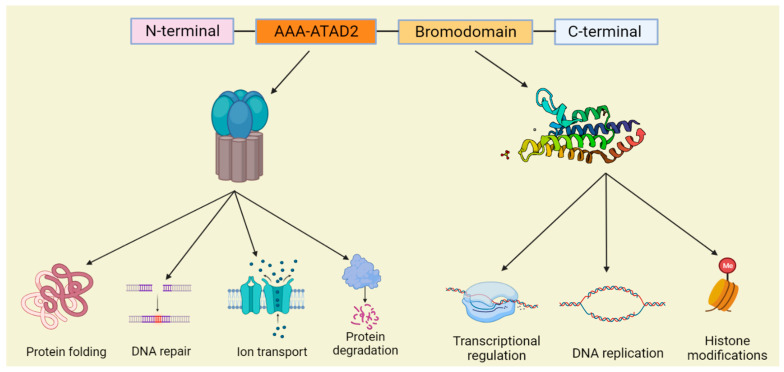
Schematic diagram of the ATAD2 domain structure and the partial functions of the AAA-ATAD2 and bromodomain.

**Figure 2 biomolecules-12-01040-f002:**
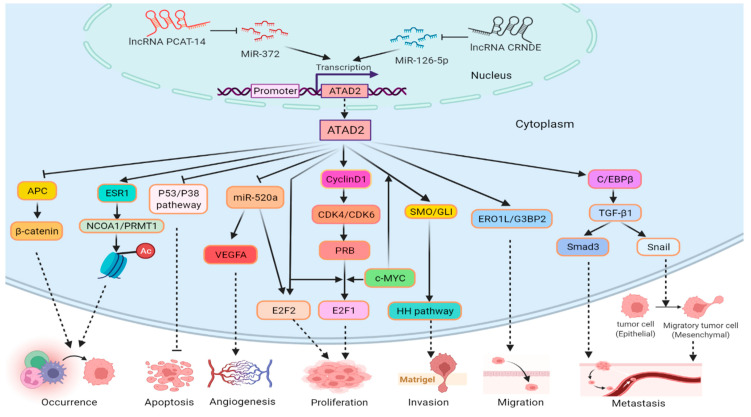
Role of ATAD2 in digestive system malignant tumors. lncRNA PCAT-14 and CRND inhibit, respectively, miR-372 and miR126-5p to promote the transcription of *ATAD2*. ATAD2 promotes tumor cell proliferation, invasion, migration, and tumorigenesis through the pRb-E2F1 pathway, HH pathway, ERO1L/G3BP2, and APC/β-catenin. ATAD2 increases the differential expression of NCOA1/PRMT1 via ESR1, which affects the acetylation level of histones to promote tumorigenesis. ATAD2 represses miR-520a and promotes the expression of VEGFA and E2F2, which are associated with tumor angiogenesis and tumor cell proliferation, separately. ATAD2 promotes the expression of Snail and Smad3 via C/EBPβ/ TGF-β1, Smad3 facilitates tumor metastasis, and Snail enhances it by EMT. ATAD2 also promotes proliferation and inhibits apoptosis, respectively, via the E2F2 and p53/p38 pathways in tumor cells.

**Figure 3 biomolecules-12-01040-f003:**
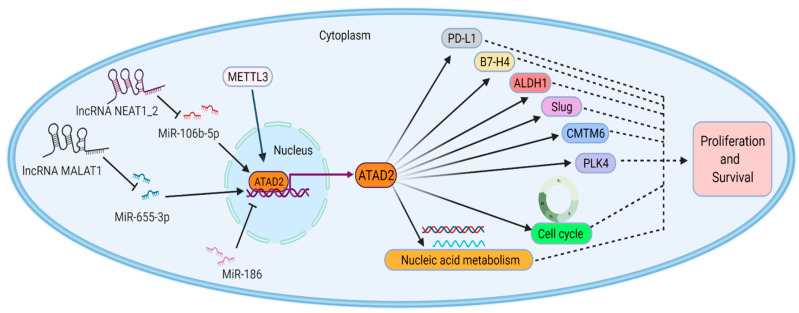
Partial roles of ATAD2 in other types of malignant tumors. lncRNA NEAT1_2 and lncRNA MALAT1 promote *ATAD2* expression by repressing miR-106b-5p and miR-655-3p, respectively. Moreover, METTL3 promotes *ATAD2* expression, but miR-186 negatively regulates *ATAD2* expression. ATAD2 promotes tumor cell proliferation and survival by regulating cell cycle, nucleic acid metabolism, PD-L1, B7-H4, ALDH1 Slug, and CMTM6 proteins.

**Table 1 biomolecules-12-01040-t001:** Expression profiles and upstream and downstream targets of ATAD2 in human malignancies.

Systems	Tumor Type	Role	Expression	Upstream Targets	Downstream Targets	References
Digestive system	EC	tumor promotor	upregulation	-	C/EBPβ, HH/WNT/NOTCH/pathway	[16,17,18,19]
	GC	tumor promotor	upregulation	HIF1α	pRb-E2F1 pathway, ESR1	[20,21,22,23]
	HCC	tumor promotor	upregulation	miR-372	miR-520a/E2F2 pathway, ERO1L/G3BP2, SMO/GLI, APC, p53/p38 pathway, KIF15	[24,25,26,27,28,29,30]
	PC	tumor promotor	upregulation	miR-217	-	[31,32]
	CRC	tumor promotor	upregulation	miR-126-5p	pRb-E2Fs pathway, TRIM25, miR-520a	[33,34,35,36]
Reproductive system	OC	tumor promotor	upregulation	miR-372, miR-200b-5p, miR-302	PI3K/AKT pathway, MAPK pathway, APC	[37,38,39,40]
	UCEC	tumor promotor	upregulation	MYC	B-MYB, E2Fs, KIFs	[41,42]
	CC	tumor promotor	upregulation	-	-	[43]
Urinary system	PCa	tumor promotor	upregulation	AR, E2F1	NSD2, AR/AR target gene	[44]
	RC	tumor promotor	upregulation	miR-372	-	[45]
Respiratory system	LC	tumor promotor	upregulation	MYC, HIF-1α	AKT-GLUT1/HK2 pathway, PI3K/AKT pathway, mtROS	[46,47,48]
Others	BC	tumor promotor	upregulation	MYC, E2, pRB/E2F pathway	MYC, MYC/ERα target gene, ACTR, KIFs, PI3K/AKT/mTOR pathway, Chk1, Chk2, BRCA1	[9,12,41,49,50,51,52,53,54]
	OS	tumor promotor	upregulation	METTL3	-	[55]
	RB	tumor promotor	upregulation	miR-186, miR-655-3p	HH pathway	[56,57]
	OSCC	tumor promotor	upregulation	-	PD-L1, B7-H4, ALDH1, Slug, CMTM6	[58]
	PTC	tumor promotor	upregulation	miR-106b-5p	-	[59]
	GBM	tumor promotor	upregulation	-	PLK4	[60]
	NPC	tumor promotor	upregulation	-	cell cycle and nucleic acid metabolism	[61]

EC esophageal cancer, GC gastric cancer, HCC hepatocellular cancer, PC pancreatic cancer, CRC colorectal cancer, OC ovarian cancer, UCEC uterine corpus endometrial carcinoma, CC cervical cancer, PCa prostate cancer, RC renal cancer, LC lung cancer, BC breast cancer, OS osteosarcoma, RB retinoblastoma, OSCC oral squamous cell carcinoma, PTC papillary thyroid cancer, GBM glioblastoma, NPC nasopharyngeal carcinoma.

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
