# Peer review of "Tumor-Promoting ATAD2 and Its Preclinical Challenges"

_biomolecules, 2022, doi:10.3390/biom12081040_

Round 1

Reviewer 1 Report

In the current manuscript, Liu et al. attempted to summarize the current basic science and challenges of clinical development to target the tumor-promoting gene ATAD2. Overall, the manuscript is fairly well-written and well-sourced. However, the layout of the review is somewhat repetitive and uninspiring.

Below are my comments for the current manuscript.

1.     Instead of going over the roles of ATAD2 in every single type of cancer with some cancers with very little information, the authors should re-organize the review to highlight the roles of ATAD2 in different oncogenic pathways and incorporate the importance of ATAD2 in each cancer type through the oncogenic pathways discussed.

2.     The authors should provide a schematic diagram outlining the domain structure of ATAD2 with each domain and its function (AAA domain, BRD domain, etc.). Also, it would be useful to compare the ATAD2 domain structure to closely related proteins within the same protein family.

3.     In Table 1, in the second column where the authors listed all the tumor types (EC, GC, HCC, PC, etc.), it would be helpful to include the key for the abbreviation right underneath the table so that the readers can know what cancer it is.

4.     The authors should incorporate the following up-to-date and important studies about ATAD2:

a.     Nayak, A., Kumar, S., Singh, S. P., Bhattacharyya, A., Dixit, A., & Roychowdhury, A. (2021). Oncogenic potential of ATAD2 in stomach cancer and insights into the protein-protein interactions at its AAA + ATPase domain and bromodomain. Journal of biomolecular structure & dynamics, 1–17. Advance online publication. https://doi.org/10.1080/07391102.2021.1871959

b.     Baggiolini, A., Callahan, S. J., Montal, E., Weiss, J. M., Trieu, T., Tagore, M. M., Tischfield, S. E., Walsh, R. M., Suresh, S., Fan, Y., Campbell, N. R., Perlee, S. C., Saurat, N., Hunter, M. V., Simon-Vermot, T., Huang, T. H., Ma, Y., Hollmann, T., Tickoo, S. K., Taylor, B. S., … White, R. M. (2021). Developmental chromatin programs determine oncogenic competence in melanoma. Science (New York, N.Y.)373(6559), eabc1048. https://doi.org/10.1126/science.abc1048

c.     Wang, T., Perazza, D., Boussouar, F., Cattaneo, M., Bougdour, A., Chuffart, F., Barral, S., Vargas, A., Liakopoulou, A., Puthier, D., Bargier, L., Morozumi, Y., Jamshidikia, M., Garcia-Saez, I., Petosa, C., Rousseaux, S., Verdel, A., & Khochbin, S. (2021). ATAD2 controls chromatin-bound HIRA turnover. Life science alliance4(12), e202101151. https://doi.org/10.26508/lsa.202101151

d.     Ekin, U., Yuzugullu, H., Ozen, C., Korhan, P., Bagirsakci, E., Yilmaz, F., Yuzugullu, O. G., Uzuner, H., Alotaibi, H., Kirmizibayrak, P. B., Atabey, N., Karakülah, G., & Ozturk, M. (2021). Evaluation of ATAD2 as a Potential Target in Hepatocellular Carcinoma. Journal of gastrointestinal cancer52(4), 1356–1369. https://doi.org/10.1007/s12029-021-00732-9

e.     Hao, S., Li, F., Jiang, P., & Gao, J. (2022). Effect of chronic intermittent hypoxia-induced HIF-1α/ATAD2 expression on lung cancer stemness. Cellular & molecular biology letters27(1), 44. https://doi.org/10.1186/s11658-022-00345-5

f.      Dutta, M., Das, B., Mohapatra, D., Behera, P., Senapati, S., & Roychowdhury, A. (2022). MicroRNA-217 modulates pancreatic cancer progression via targeting ATAD2. Life sciences301, 120592. https://doi.org/10.1016/j.lfs.2022.120592

Author Response

Dear Reviewer 1,

We thank you very much for considering our manuscript, so as to your positive comments and constructive suggestions. Accordingly, we have revised the manuscript based on your suggestion. Responses to specific comments are addressed below:

In this manuscript, the authors should re-organize the review to highlight the roles of ATAD2 in different oncogenic pathways and incorporate the importance of ATAD2 in each cancer type through the oncogenic pathways discussed, instead of going over the roles of ATAD2 in every single type of cancer with some cancers with very little information.

Response: This is a rational and logical comment. However, the role of ATAD2 varies in different tumor types. Although some of them have similar ATAD2 signaling cascades, the heterogeneity among tumors affects the function of ATAD2 as well as their clinical prospect. In addition, the oncogenic pathway of ATAD2 was well described by Nayak et al. (2021) in their article. Therefore, we would like to retain the current manuscript layout.

In this manuscript, the authors should provide a schematic diagram outlining the domain structure of ATAD2 with each domain and its function (AAA domain, BRD domain, etc.). Also, it would be useful to compare the ATAD2 domain structure to closely related proteins within the same protein family.

Response: We thank for this suggestion. We have added a schematic diagram in the manuscript to provide a brief overview of the domain structural of ATAD2 and its function and compare ATAD2 with the homologous gene ATAD2B in section 2.

In this manuscript, the authors should add the key for the abbreviation right underneath the table so that the readers can know what cancer it is.

Response: We thank for this comment. We have added a footnote at the bottom of Table 1.

In this manuscript, the authors should incorporate the following up-to-date and important studies about ATAD2: 

Nayak, A., Kumar, S., Singh, S. P., Bhattacharyya, A., Dixit, A., &

Roychowdhury, A. (2021). Oncogenic potential of ATAD2 in stomach cancer and insights into the protein-protein interactions at its AAA + ATPase domain and bromodomain. Journal of biomolecular structure & dynamics, 1–17. Advance online publication. https://doi.org/10.1080/07391102.2021.1871959

Baggiolini, A., Callahan, S. J., Montal, E., Weiss, J. M., Trieu, T., Tagore, M.

M., Tischfield, S. E., Walsh, R. M., Suresh, S., Fan, Y., Campbell, N. R., Perlee, S. C., Saurat, N., Hunter, M. V., Simon-Vermot, T., Huang, T. H., Ma, Y., Hollmann, T., Tickoo, S. K., Taylor, B. S., … White, R. M. (2021). Developmental chromatin programs determine oncogenic competence in melanoma. Science (New York, N.Y.), 373(6559), eabc1048. https://doi.org/10.1126/science.abc1048

Wang, T., Perazza, D., Boussouar, F., Cattaneo, M., Bougdour, A., Chuffart,

F., Barral, S., Vargas, A., Liakopoulou, A., Puthier, D., Bargier, L., Morozumi, Y., Jamshidikia, M., Garcia-Saez, I., Petosa, C., Rousseaux, S., Verdel, A., & Khochbin, S. (2021). ATAD2 controls chromatin-bound HIRA turnover. Life science alliance, 4(12), e202101151. https://doi.org/10.26508/lsa.202101151

Ekin, U., Yuzugullu, H., Ozen, C., Korhan, P., Bagirsakci, E., Yilmaz, F., Yuzugullu, O. G., Uzuner, H., Alotaibi, H., Kirmizibayrak, P. B., Atabey, N., Karakülah, G., & Ozturk, M. (2021). Evaluation of ATAD2 as a Potential Target in Hepatocellular Carcinoma. Journal of gastrointestinal cancer, 52(4), 1356–1369. https://doi.org/10.1007/s12029-021-00732-9

Hao, S., Li, F., Jiang, P., & Gao, J. (2022). Effect of chronic intermittent hypoxia-induced HIF-1α/ATAD2 expression on lung cancer stemness. Cellular & molecular biology letters, 27(1), 44. https://doi.org/10.1186/s11658-022-00345-5

Dutta, M., Das, B., Mohapatra, D., Behera, P., Senapati, S., & Roychowdhury, A. (2022). MicroRNA-217 modulates pancreatic cancer progression via targeting ATAD2. Life sciences, 301, 120592. https://doi.org/10.1016/j.lfs.2022.120592

Response: We thank for this suggestion. We have incorporated the above six up-to-date and important studies about ATAD2, as follows.

Incorporated the study of Nayak et al. into the gastric cancer segment of section 4.

Incorporated the study of Baggiolini et al. into the introduction segment of section 1.

Incorporated the study of Wang et al. into the “other functions” segment of Section 3.

Incorporated the study of Ekin et al. into the hepatocellular cancer segment of section 4.

Incorporated the study of Hao et al. into the lung cancer segment of section 4.

Incorporated the study of Dutta et al. into the pancreatic cancer segment of section 4.

Thanks again for your positive comments and encouragement.

Yours Sincerely,

Haicheng Liu

Reviewer 2 Report

The present review by Liu H et al. summarizes and presents well the available information about the ATAD2 pathway. This is essential and potentially crucial for anticancer drugs and presents similar characteristics to KRAS. I would recommend authors to include a footnote in Table 1 to clarify the abbreviations for tumor type, as they are given later in the text. Also, I am unsure if the classification of breast cancer as a reproductive system cancer is correct, and authors may consider moving it to others. Finally, the authors should enhance section 5, “Preclinical challenges for ATAD2,” and describe in more detail the available preclinical studies for the developed ATAD2 inhibitors.

Author Response

Dear Reviewer 2,

We thank you very much for considering our manuscript, so as to your positive comments and constructive suggestions. Accordingly, we have revised the manuscript based on your suggestion. Responses to specific comments are addressed below:

In this manuscript, the authors should add a footnote in Table 1 to clarify the abbreviations for tumor type, as they are given later in the text.

Response: We thank for this comment. We have added a footnote at the bottom of Table 1.

In this manuscript, the authors should consider if the classification of breast cancer as a reproductive system cancer is correct, and authors may consider moving it to others.

Response: We thank for this suggestion.  We have moved breast cancer to other types of malignant tumors.

In this manuscript, the authors should enhance section 5, “Preclinical challenges for ATAD2,” and describe in more detail the available preclinical studies for the developed ATAD2 inhibitors.

Response: We thank for this comment. We have expanded section 5 to focus on the latest research on ATAD2 inhibitors.

Thanks again for your positive comments and encouragement.

Yours Sincerely,

Haicheng Liu

Round 2

Reviewer 1 Report

The authors have adequately addressed all of my concerns. I only have one small request left. 

Since this is a review article, the authors should bold, highlight, and annotate some articles in the reference section that the authors think are important and of high interest to the readers. This is routinely done for review articles in high impact journal.